Subject Areas:
civil engineering/materials science

Keywords:
modified recycled aggregate concrete, compressive strength, damping property, interfacial transition zone, rubber powder, steel fibre

Author for correspondence:
Zhimin Yao
e-mail: yaozm@whut.edu.cn

# Experimental study on the compressive strength, damping and interfacial transition zone properties of modified recycled aggregate concrete

Bin Lei[1,2], Huajian Liu[1], Zhimin Yao[2,3] and Zhuo Tang[2]

[1]School of Civil Engineering and Architecture, Nanchang University, Nanchang, Jiangxi 330031, People's Republic of China
[2]Center for Built Infrastructure Research (CBIR), School of Civil and Environmental Engineering, University of Technology Sydney, Sydney, New South Wales 2007, Australia
[3]School of Energy and Power Engineering, Wuhan University of Technology, Wuhan 430063, People's Republic of China

ZY, 0000-0002-3060-9023

At present, many modification methods have been proposed to improve the performance of recycled aggregate concrete (RAC). In this study, tests on the compressive strength and damping properties of modified RAC with the addition of different proportions of recycled coarse aggregate (RCA) (0, 50, 100%), rubber powder (10, 15, 20%), steel fibre (5, 7.5, 10%) and fly ash (15, 20, 5%) are carried out. To elucidate the effect of the modification method on the interfacial transition zone (ITZ) performance of RAC, model ITZ specimens are used for push-out tests. The results show that when the replacement rate of RCA reaches 100%, the loss factor of the RAC is 6.0% higher than that of natural aggregate concrete; however, the compressive strength of the RAC decreases by 22.6%. With the addition of 20% rubber powder, the damping capacity of the modified RAC increases by 213.7%, while the compressive strength of the modified RAC decreases by 47.5%. However, with the addition of steel fibre and fly ash, both the compressive strength and loss factor of the RAC specimens increase. With a steel fibre content of 10 wt%, the compressive strength and loss factor of the RAC increase by 21.9% and 15.2%, respectively. With a fly ash content of 25 wt%, the compressive strength and loss factor of the RAC increase by 8.6% and 6.9%, respectively. This demonstrates that steel fibre and fly ash are effective in improving both the damping properties and compressive

strength of RAC, and steel fibre is more effective than fly ash. Two methods were used for modification of the RAC: reinforcing the RCA through impregnation with a 0.5% polyvinyl alcohol (PVA) emulsion and nano-SiO$_2$ solution, and strengthening the RAC integrally through the addition of fly ash as an admixture. Both of these techniques can improve the ITZ bond strength between the RAC and new mortar. Replacing 10% of the cement with fly ash in the new mortar is shown to be the best method to improve the ITZ strength.

## 1. Introduction

With the growing interest in sustainable and environmentally friendly construction, recycled aggregate concrete (RAC) has been widely investigated and is gradually being incorporated into real engineering projects. However, many studies have found that the mechanical properties of RAC decrease with an increasing replacement rate of recycled coarse aggregate (RCA) [1–3]. The compressive strength of RAC is generally 10–30% lower than that of natural aggregate concrete (NAC) [4–6], while the flexural strength is generally approximately 10% lower than that of NAC [5]. The tensile strength of RAC is generally approximately 6% lower than that of NAC, and the elastic modulus is generally 10–33% lower than that of NAC [5,7]. As a result, a considerable number of remedial measures have been proposed to improve both the mechanical characteristics and the durability of RAC. The addition of admixtures, fibres and aggregate impregnation has been investigated.

Various methods for adding an admixture have been investigated to improve the strength of RAC. The influence of the rubber content and particle size on the compressive strength, elastic modulus, peak and ultimate strains, crack characteristics and failure mode of RAC has been investigated [8,9]. Aslani *et al.* [10] proposed a method for producing self-compacting concrete through the utilization of RCA and recycled crumb rubber; the fresh properties as well as the compressive and tensile strengths were determined to evaluate the optimal mix design of RAC.

The addition of steel fibre can also significantly improve the compressive strength and alter the fracture process of RAC [11–13]. Furthermore, experimental results have indicated that steel fibre also has a positive effect on the flexural performance and shear properties of reinforced RAC [14,15]. Afroughsabet *et al.* [12] found that the splitting tensile strength and flexural strength of steel fibre-reinforced RAC increased by 60 and 88% after adding 1% double hook-end steel fibres to RAC in 28 days.

Fly ash is an industrial solid waste that can be used as a replacement for cement in the manufacture of concrete. Kim *et al.* [16] concluded that the replacement of cement with fly ash results in little reduction in the compressive strength of RAC, and the positive effect on the chloride resistance is more significant. However, it appears that a high fly ash content can reduce the performance of RAC, which means the improvement achievable by incorporating fly ash is limited. For example, Eva *et al.* [17] and Kou & Poon [18] recommend 25% fly ash as a replacement for cement in concrete made from RCA to achieve desirable properties.

The treatment of RCA with nanomaterial solutions or polymer emulsions has also been investigated to strengthen RCA [19,20]. When RCA is immersed in a solution of nanomaterials such as nano-SiO$_2$ or nano-CaCO$_3$, the nanomaterials can fill the pores and voids inside the adhered mortar and then react with calcium hydroxide to form (C–S–H) gels; thus, enhancing the strength and durability of RAC [21]. Polymers are water-repellent and can be used to reduce the water absorption of porous materials. When RCA is immersed in a polymer emulsion, the polymer molecules can fill the pores of the adhered mortar and seal the surfaces of the RCA [19,22,23].

Damping is an important dynamic property of concrete structures. A number of studies have been conducted on modifying the molecular structure of concrete in order to enhance the internal damping without negatively affecting other properties such as mechanical properties and durability [24]. Skripkiunas *et al.* [25] found that replacing fine aggregate with waste rubber could significantly increase the damping capacity of concrete. Yan *et al.* [26] found that crimped-fibre or fine-fibre reinforced concrete also has an improved damping capacity. However, there are few studies on the damping properties of modified RAC. Liang *et al.* [27] indicated that both increasing the substitution rate and decreasing the size of RCA can lead to an increase in the damping ratio, which may be induced by sliding in the interfacial transition zone (ITZ) or an increase in micro- and macro-cracks. Li *et al.* [28] concluded that the damping variation in RAC is mainly due to the new ITZ and old ITZ.

Therefore, in this study, a series of experiments are conducted on the compressive strength and damping properties of modified RAC with three types of admixtures: rubber powder, steel fibre and

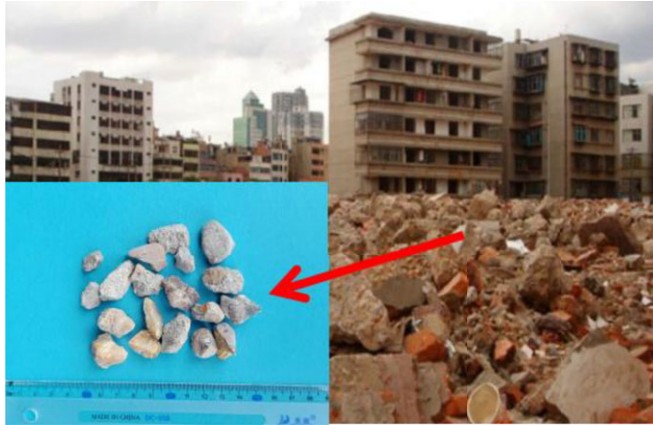

**Figure 1.** Recycled coarse aggregate.

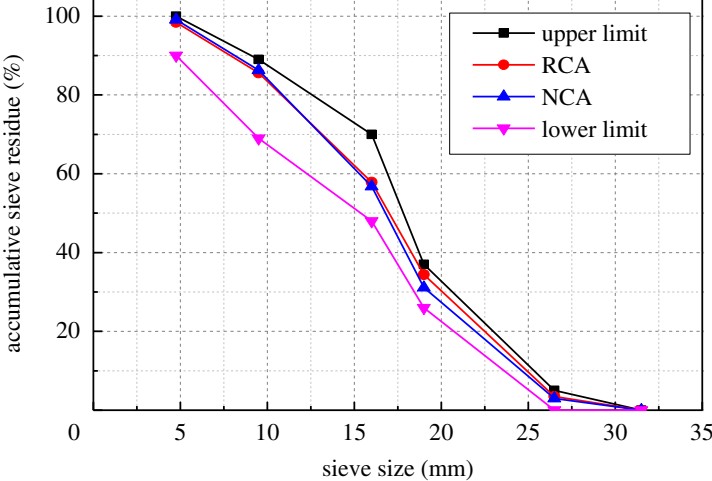

**Figure 2** Gradation curves for RCA and NCA.

fly ash. Moreover, the interfacial properties between the RCA and new mortar in the ITZ of the specimens are also investigated with the push-out test method. The results of this study can be used as a theoretical basis for future related studies and as a reference for the promotion and application of RAC structures.

# 2. Methodology

## 2.1. Raw materials

The RCA in this study originated from the demolition of an old building in Nanchang city, China. After crushing and screening, RCA that meets the requirements for particle size and roughness can be obtained (figure 1). The gradation curves of RCA and NCA are shown in figure 2 and the physical properties of RCA and NCA are summarized in table 1.

Sand used as fine aggregate was collected from the Ganjiang River in Jiangxi Province, China; it has an apparent density of 2487.9 kg m$^{-3}$. The ordinary Portland cement is of the type P.O.42.5 produced by HaiLuo Ltd, with a final setting time of 237 min. The water used for the experiments was tap water obtained from the laboratory. In this study, #20 rubber powder, corrugated steel fibres and fly ash were used in the modified RAC, as shown in figure 3 and a commercial nano-SiO$_2$ solution was used to treat RCA at a content of 30%. The basic properties of the Portland cement, rubber powder, steel fibre, fly ash and polyvinyl alcohol powder are summarized in tables 2–5. To prepare the required polyvinyl alcohol immersion solution for ITZ modelling of the specimens, polyvinyl alcohol powder is slowly added into cold water at about 20°C while stirring to prevent powder caking. The stirring

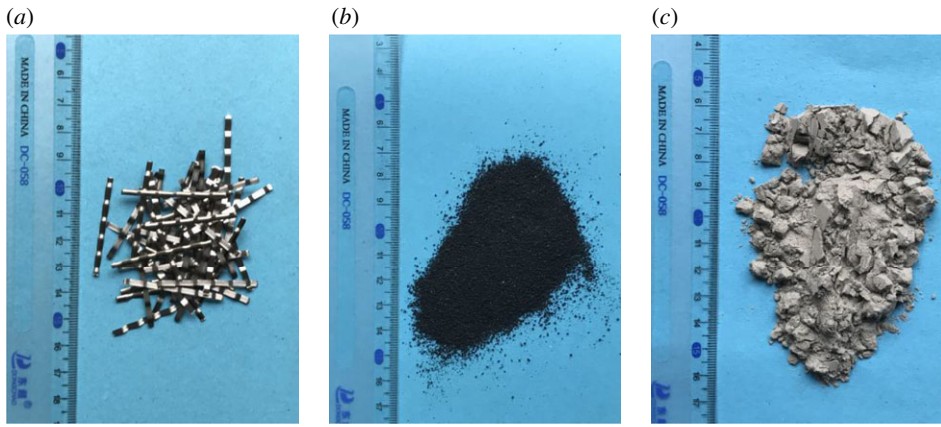

**Figure 3.** Admixture in modified RAC: (*a*) steel fibres, (*b*) rubber powder and (*c*) fly ash.

**Table 1.** Physical properties of RCA and NCA.

| type of coarse aggregate | apparent density (kg m$^{-3}$) | water absorption (%) | crushing index (%) | size distribution (mm) |
|---|---|---|---|---|
| RCA | 2610 | 3.88 | 18.6 | 5–31.5 |
| NCA | 2726 | 0.47 | 7.7 | 5–31.5 |

**Table 2.** Chemical composition (%) of Portland cement and fly ash.

| raw materials | SiO$_2$ | Al$_2$O$_3$ | CaO | MgO | Fe$_2$O$_3$ | SO$_3$ | K$_2$O | Na$_2$O | total |
|---|---|---|---|---|---|---|---|---|---|
| Portland cement | 21.92 | 7.63 | 60.1 | 2.59 | 2.60 | 2.53 | 0.26 | 0.32 | 97.99 |
| fly ash | 56.10 | 26.24 | 5.57 | 1.20 | 5.34 | 1.21 | 1.93 | 0.28 | 97.87 |

**Table 3.** Composition (%) of the rubber powder.

| rubber hydrocarbon | carbon black | acetone | ash content | sulfur |
|---|---|---|---|---|
| 52.3 | 30.7 | 10.7 | 3.2 | 1.4 |

**Table 4.** Physical properties of the steel fibre.

| apparent density (g cm$^{-3}$) | length (mm) | diameter (mm) | tensile strength (MPa) | elastic modulus (GPa) |
|---|---|---|---|---|
| 7.8 | 30 | 0.5 | 1132 | 200 |

**Table 5.** Composition (%) of the polyvinyl alcohol powder.

| polyvinyl alcohol | volatiles | sodium acetate | ash content |
|---|---|---|---|
| 91.7 | 5.0 | 2.8 | 0.5 |

speed is 70–100 r.p.m. Impregnation treatment of the RCA was achieved by soaking the aggregates in the immersion solution. First, solutions of the polyvinyl alcohol emulsion and nano-silicon with good dispersion were prepared at the required concentrations. The RCA was then added to the bath and soaked for 120 min. The RCA was then removed from the bath, and a screen was used to remove the

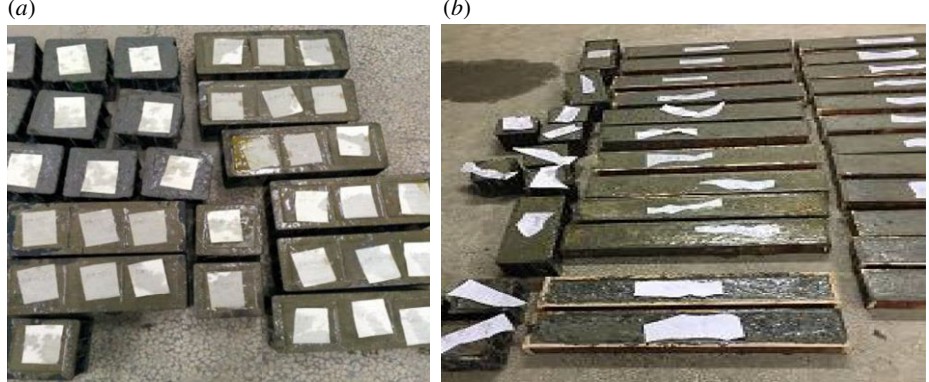

**Figure 4.** RAC test specimens: (*a*) samples for the compressive strength tests and (*b*) beams for the loss factor tests.

excess polyvinyl alcohol and nano-silicon adhering to the RCA. Finally, the RCA was dried at a temperature of $20 \pm 2$°C and a relative humidity of $64\% \pm 5\%$ for at least 3 days.

## 2.2. Preparation of RCA

In the tests, the substitution rate of RCA for NCA is 0%, 50% and 100%, denoted as RC-0-0, RC-0-50% and RC-0-100%, respectively. For the RAC modified with rubber powder, steel fibre and fly ash, 100% RCA is used in each group. For the recycled concrete modified with rubber powder, the percentage of cement replaced with rubber powder is 10%, 15% and 20% by mass, which are denoted as RC-1-10, RC-1-15 and RC-1-20, respectively. For the RAC modified with steel fibre, the percentage of cement replaced with steel fibre is 5%, 7.5% and 10%, which are denoted as RC-2-5, RC-2-7.5 and RC-2-10, respectively. For RAC modified with fly ash, the percentage of cement replaced by fly ash is 15%, 20% and 25%, denoted as RC-3-15, RC-3-20 and RC-3-25, respectively. Three cubes and three corresponding beams were manufactured for each group to measure the compressive strength and damping properties of the concrete. The concrete cubes and beams have sizes of $100 \times 100 \times 100$ mm and $100 \times 50 \times 650$ mm, respectively, as shown in figure 4. The mixture proportions of the concrete in each group are summarized in table 6.

## 2.3. Damping test specimens

The simply supported beam method is used to measure the damping properties of RAC [29]. The beam is first excited by free vibration at the midpoint of the RAC beam, and the response signal is recorded by the installed sensor and used to calculate the damping loss factor. The damping loss factor, $\eta$, is calculated with the below equation

$$\eta = \frac{f_H - f_L}{f_N},\tag{2.1}$$

where $\eta$ is the loss factor of the RAC; $f_H$ and $f_L$ are the frequencies (Hz) with an amplitude increase and decrease of 3 dB, respectively; $f_N$ is the resonance frequency (Hz) of the $N$th step, which always chooses the value of the first step.

## 2.4. ITZ model specimens

In this study, push-out tests are used to study the effect of the modification method on the ITZ performance between the RCA and new mortar. The ITZ model specimens used for the push-out tests are composed of two parts: the mortar matrix and cylindrical RCA (figure 5). The cylindrical RCA is obtained by coring beams in old building structures. The proportions of the new mortar are cement : sand : water = 2.04 : 2.8 : 1. Four types of ITZ model specimens were considered: Group A are composed of new cement mortar and untreated cylindrical RCA; Group B are new cement mortar and cylindrical RCA soaked in 0.5% nano-SiO$_2$; Group C are new cement mortar and cylindrical RCA impregnated in a 0.5% polyvinyl alcohol solution and Group D are new mortar with fly ash replacing 10% of the cement and untreated RCA. The experimental device is designed by referring to Sinan Caliskan [30], and the push-out test device is shown in figure 6.

**Table 6.** Mix proportions of the concrete for each group.

| specimen number | cement (kg m$^{-3}$) | sand (kg m$^{-3}$) | RCA (Kg m$^{-3}$) | NCA (kg m$^{-3}$) | water (kg m$^{-3}$) | rubber powder (kg m$^{-3}$) | steel fibre (kg m$^{-3}$) | Fly ash (kg m$^{-3}$) | 28 days compressive strength (MPa) |
|---|---|---|---|---|---|---|---|---|---|
| RC-0-0 | 398 | 546 | — | 1160 | 195 | 0 | 0 | 0 | 38.9 |
| RC-0-50 | 398 | 546 | 580 | 580 | 195 | 0 | 0 | 0 | 34.8 |
| RC-0-100 | 398 | 546 | 1160 | 0 | 195 | 0 | 0 | 0 | 30.1 |
| RC-1-10 | 358.2 | 546 | 1160 | 0 | 195 | 39.8 | 0 | 0 | 22.3 |
| RC-1-15 | 338.3 | 546 | 1160 | 0 | 195 | 59.7 | 0 | 0 | 20.4 |
| RC-1-20 | 318.4 | 546 | 1160 | 0 | 195 | 79.6 | 0 | 0 | 15.8 |
| RC-2-5 | 378.1 | 546 | 1160 | 0 | 195 | 0 | 19.9 | 0 | 32.6 |
| RC-2-7.5 | 368.1 | 546 | 1160 | 0 | 195 | 0 | 29.9 | 0 | 34.5 |
| RC-2-10 | 358.2 | 546 | 1160 | 0 | 195 | 0 | 39.8 | 0 | 36.7 |
| RC-3-15 | 338.3 | 546 | 1160 | 0 | 195 | 0 | 0 | 59.7 | 29.9 |
| RC-3-20 | 318.4 | 546 | 1160 | 0 | 195 | 0 | 0 | 79.6 | 30.7 |
| RC-3-25 | 299.2 | 546 | 1160 | 0 | 195 | 0 | 0 | 98.8 | 32.9 |

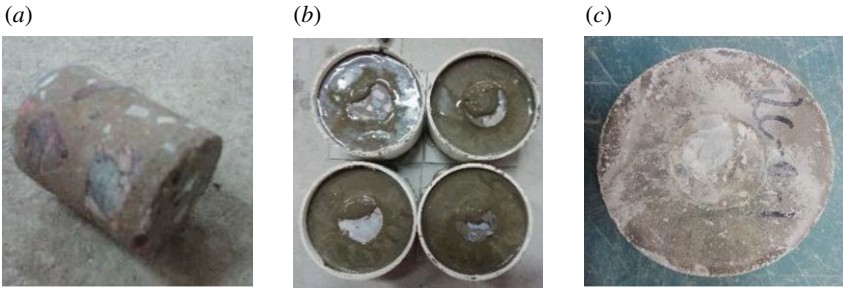

**Figure 5.** Casting process of the ITZ model specimens: (*a*) cylindrical RCA, (*b*) placement of the cylindrical RCA and pouring new mortar in the test mould and (*c*) the ITZ model specimen after demoulding.

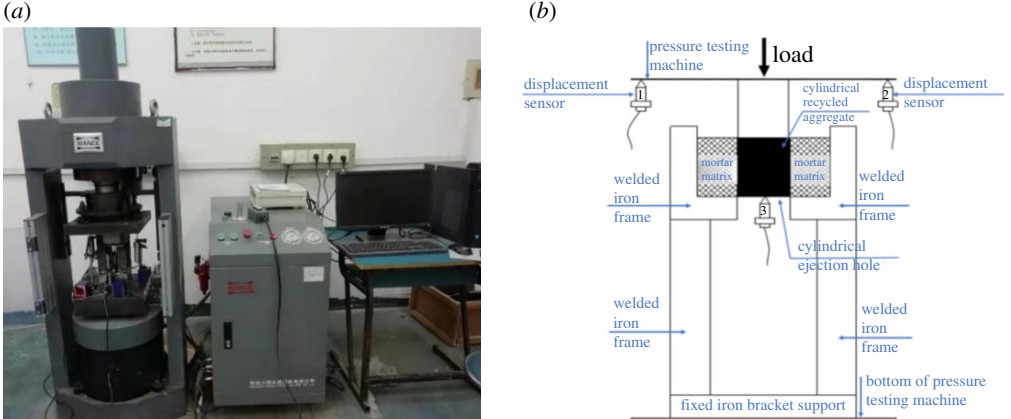

**Figure 6.** Testing device for the ITZ model specimens: (*a*) testing device and (*b*) schematic diagram of the testing device.

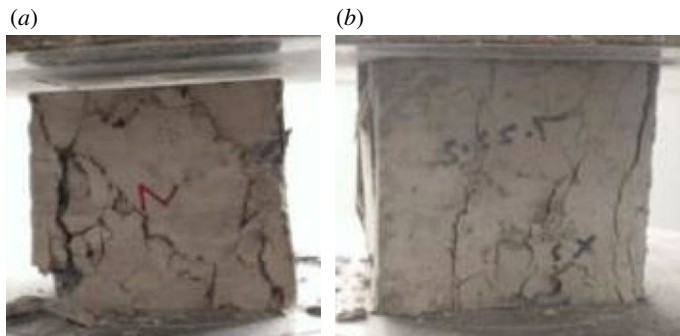

**Figure 7.** Compressive failure mode of samples: (*a*) RAC and (*b*) NAC.

# 3. Results and discussion

## 3.1. Damping and compressive strength

### 3.1.1. RCA substitution rate

The failure modes in the compression tests of the natural concrete cube (RC-0-0) and the RAC cube with 100% RCA (RC-0-100) are shown in figure 7. For both the natural and recycled samples, with increasing external load, oblique cracks first appeared on both sides of the concrete block and gradually spread to the centre. Meanwhile, there was spalling of the mortar around the recycled and natural concrete cube, as shown in figure 7*a*,*b*, respectively. It appears that the RAC experienced a more severe spalling phenomenon than the NAC.

The influence of the substitution of NCA with RCA on the compressive strength and loss factor of the RAC is shown in figure 8. As shown in figure 8, the loss factor increases with an increasing replacement ratio, from 5.2% (RC-0-0) to 5.4% (RC-0-50) and 5.5% (RC-0-100). The loss factor is increased by 5.8% in the RAC

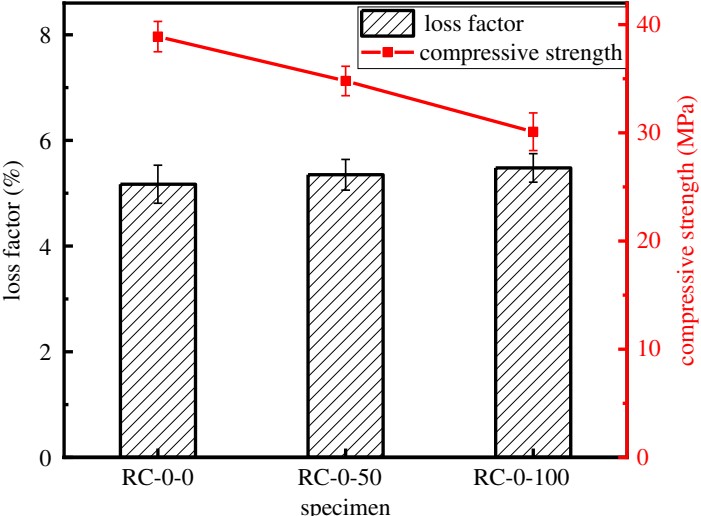

**Figure 8.** Loss factor and 28 days compressive strength of specimens with different replacements ratios with recycled aggregate.

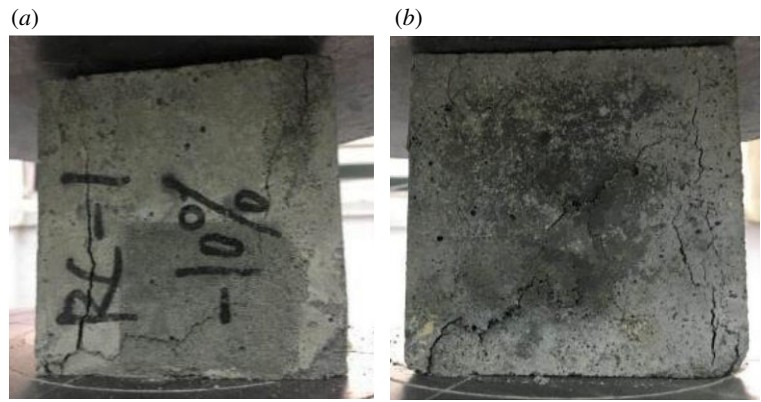

**Figure 9.** Compressive failure modes of samples with rubber powder: (*a*) modified RAC with 10% rubber powder and (*b*) modified RAC with 20% rubber powder.

with 100% replacement with RCA. However, the compressive strength of RAC decreases with increasing substitution rate. The 28 days strength of RAC decreases by 11.5% to 34.8 MPa when the substitution rate is 50%, and this decrease increases to 22.6% when the substitution rate is 100%. The increase in the loss factor is mainly due to the layer of old mortar around the RCA, which has high porosity, high water absorption and low density. In general, the initial micro-cracks internal to the RAC are increased, and thus reduce the compression properties of RAC. Moreover, when vibration forces are applied to the interface of the RCA, slip deformation occurs in the ITZ and the internal friction between cracks increases the consumption of damping energy. The higher the substitution rate with RCA, the more slip deformation will occur in the cracks, and more internal friction will consume damping energy [27].

### 3.1.2. Rubber powder

The typical failure behaviours during the compression test of the RAC cube with 10% rubber powder (RC-1-10) and 20% rubber powder (RC-1-20) are shown in figure 9. With increasing compression load, cracks first appeared at the boundary surface between the rubber particles and the mortar. After that the cracks spread swiftly to the bottom of the concrete and soon extended to the centre of the test block. The increasing pressure caused the cracks to become connected and form a large fracture, leading to spalling of the concrete mortar. The cracks could be generated much more easily and rapidly owing to the large difference between the elastic modulus of the rubber powder and that of the concrete. Under loading, the rubber has better deformability performance than the concrete, and the deformation variation in the conjoined materials causes severe stress concentration on the boundaries, which finally destroys the concrete cube. Moreover, there is no effective cohesive force between the rubber powder and cement paste, which also leads to the initial cracking of the concrete.

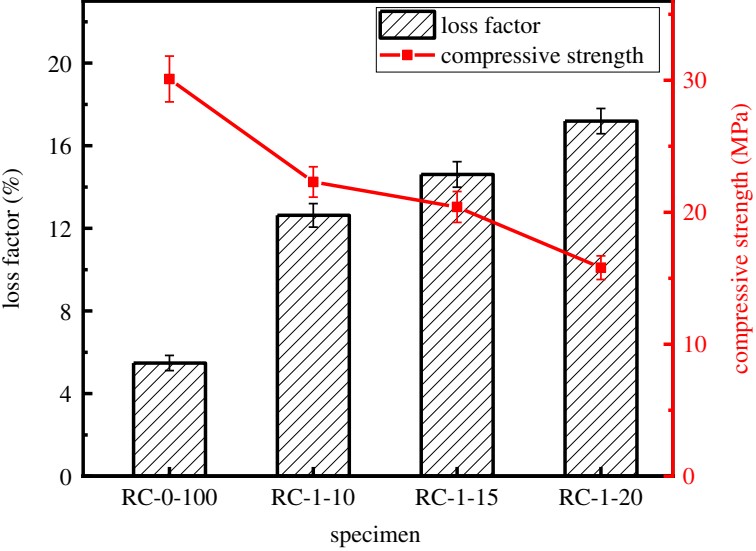

**Figure 10.** Loss factor and 28 days compressive strength of specimens with different rubber powder contents.

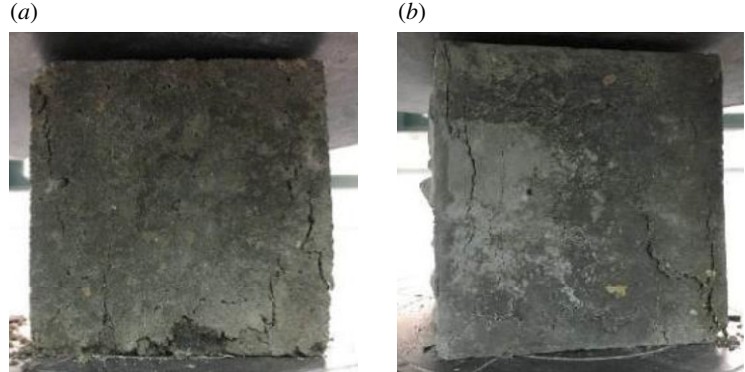

**Figure 11.** Compressive failure modes of samples with steel fibre: (*a*) modified RAC with 5% steel fibre and (*b*) modified RAC with 10% steel fibre.

The effect of varying rubber powder contents on the damping loss factor and compressive strength of the modified RAC is shown in figure 10. The damping loss factors of the modified RAC are significantly improved by the addition of rubber powder. The loss factors of the rubber powder-modified concretes are increased by 130.5% (RC-1-10), 166.6% (RC-1-15) and 213.7% (RC-1-20) compared with the reference RAC. The value of the damping loss factor is 5.5%, which rockets to almost 17.2% with a 20 wt% rubber powder content. However, with the addition of 10, 15 and 20 wt% rubber powder, the compressive strength of the modified RAC is dramatically decreased by 26.1% (RC-1-10), 32.2% (RC-1-15) and 47.5% (RC-1-20), respectively. The compressive strength with 20 wt% rubber powder is only 15.8 MPa.

As for the specific reasons for the decreased compressive strength and increased loss factor in rubber powder-modified RAC, the surface of the rubber powder is much smoother than that of the cement matrix, and thus, the bonding strength with the cement mortar is relatively weak. This causes the mortar to become more easily unattached and results in stress concentration in the location of the detached mortar, which ultimately leads to the decreased compressive strength of RAC. In addition, the elastic modulus of the rubber powder is much higher than that of the cement matrix, and thus more energy is consumed because of the active vibration and deformation of the rubber powder, which contributes to the higher loss factor in the modified RAC [25].

### 3.1.3. Steel fibre

Figure 11 shows the typical failure behaviour of RAC cubes with 5 wt% steel fibre (RC-2-5) and 10 wt% steel fibre (RC-2-10) under compressive loading. It is clear that the steel fibre has a significant positive effect on controlling the cracks. Regardless of the added steel fibre content, the number of cracks formed during the

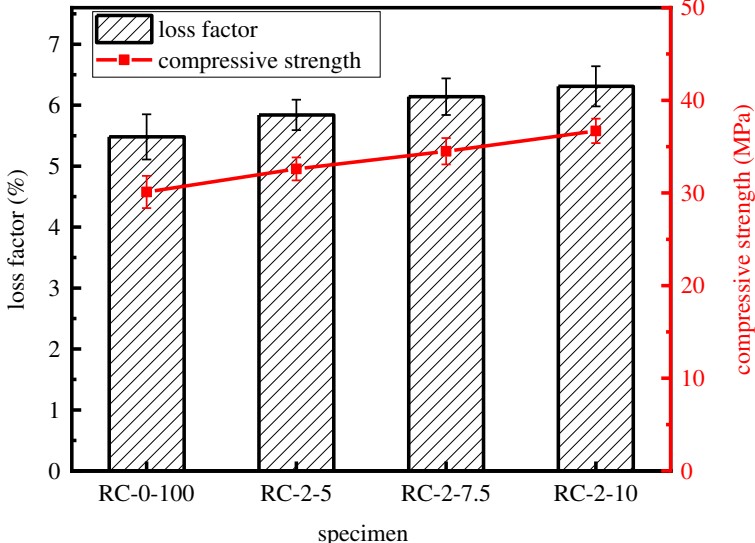

**Figure 12.** Loss factor and 28 days compressive strength of specimens with different contents of steel fibre.

compression tests is greatly reduced and their lengths are also shortened. In general, with increasing compressive loads, small cracks first appear in the steel fibre-modified RAC in the upper left corner of the concrete cube, and then gradually spread to the bottom and right sides. Furthermore, it is worth mentioning that although the cracks still increase with increasing applied loads, they remain unconnected from each other, which is normally attributed to they being blocked by the irregular distribution of the steel fibres.

The effect of different proportions of steel fibre on the compressive strength and loss factor of the modified RAC is shown in figure 12. It is notable that the 28 days compression strength of steel fibre-modified RAC is dramatically increased with the different amounts of steel fibre. Compared with the reference RAC, the compressive strength is increased by 8.3%, 14.6% and 21.9% with the addition of steel fibre at 5 wt% (RC-2-5), 7.5 wt% (RC-2-7.5) and 10 wt% (RC-2-10), respectively. Meanwhile, the loss factor of the modified RAC is also improved with increasing content of steel fibre, with increases of 6.6%, 12.1% and 15.2% for samples RC-2-5, RC-2-7.5 and RC-2-10, respectively.

The reason for the observed improvement in the compressive strength is mainly because the presence of steel fibres in the RAC alters the failure mode of the RAC. During the process of gradual deformation and failure of modified RAC with increasing compressive loads, the disorderly short steel fibres in the RAC change the path of the cracks, which hinders the development and connection of the internal cracks; thus, improving the compressive strength of the RAC. As for the increasing loss factor with increasing steel fibre content, because the stiffness of the steel fibres is greater than that of the other ingredients, more energy can be consumed during the vibration process. Furthermore, axial interfacial bonding forces are generated at the boundary of the cement mortar and steel fibre, leading to increased energy absorption and consumption during vibration, which also leads to an increase in the damping property to some extent [26].

### 3.1.4. Fly ash

The failure behaviours of the RAC cubes with 15% (RC-3-15) and 25% fly ash (RC-3-25) under compression loads are shown in figure 13. It appears that the fly ash-modified RAC exhibits a similar failure behaviour to the normal concrete (RC-0-0). Inclined cracks first appeared on both sides of the concrete block, and then spread to the centre and bottom of the concrete. As shown in figure 13, for the modified RAC with 25 wt% fly ash, the number and size of the cracks are both reduced compared with the modified RAC with 15 wt% fly ash. In other words, the fly ash could restrict the early crack development of the modified RAC and increasing the amount of fly ash is also beneficial to the RAC on a certain scale.

Figure 14 shows the compressive strength and loss factor of modified RAC with different contents of fly ash. It is interesting to note that when the fly ash content is 15%, the 28 days compressive strength of the modified RAC slightly decreases by 0.7% to 29.9 MPa. However, a different tendency is observed for fly ash contents of 20 and 25 wt% in the modified RAC, with the former compressive strength increasing slightly by 2.0% to 30.7 MPa, and the latter increasing moderately by 8.6% to

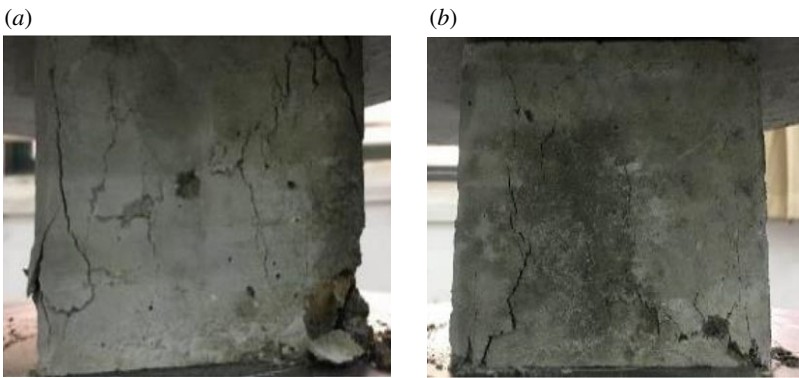

**Figure 13.** Compressive failure modes of samples with fly ash: (*a*) modified RAC with 15% fly ash and (*b*) modified RAC with 25% fly ash.

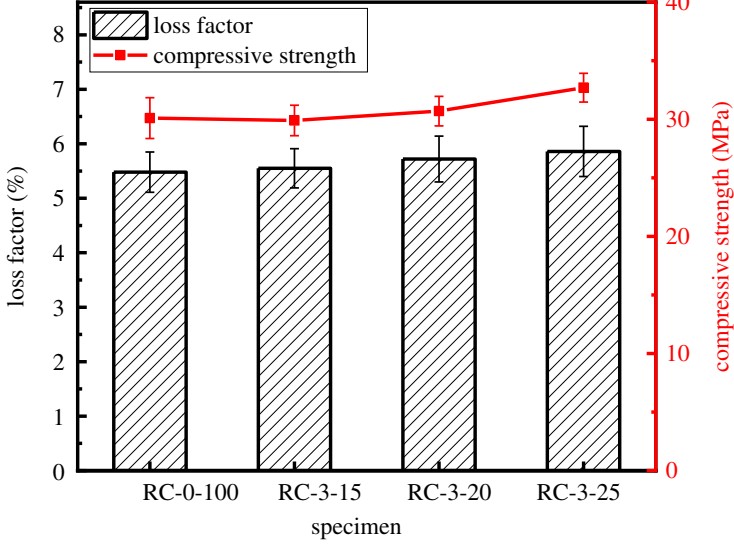

**Figure 14.** Loss factor and 28 days compressive strength of specimens with different contents of fly ash.

32.9 MPa. Similar to the performance of the steel fibre-modified RAC, the loss factor of RAC with 15, 20 and 25 wt% fly ash is correspondingly increased by 1.3%, 4.4% and 6.9%, respectively, compared with the reference RAC (RC-0-100). Fly ash seems to have no clear influence on the loss factor and the corresponding damping properties of the modified RAC.

Fly ash can react with the residual alkali in concrete to produce hydrated calcium silicate that can fill the pores inside the concrete. Furthermore, the addition of fly ash reduces the early drying shrinkage of concrete and greatly improves the density of concrete. It is generally considered that these two factors increase the compressive strength of modified RAC. Meanwhile, owing to the different vibration frequencies between the fly ash particles and cement paste, and the improved cohesion between the cement paste and RCA achieved with the addition of fly ash, the RAC will consume more energy during vibration, resulting in a loss factor that is slightly larger than that for ordinary RAC [31]. However, fly ash seems to have a minimal effect on the loss factor of the RAC, as an improvement in the loss factor of only 6.9% is achieved with a fly ash content of 25%.

## 3.2. ITZ performance of modified recycled concrete

### 3.2.1. Failure patterns of the ITZ model specimens

The cracks in the interface model specimens occur at the interface between the RCA and the mortar, and then extend to the matrix of the mortar, as shown in figure 15*a*. The mortar splits and breaks down after the push-out test. There is peeling old mortar in the RCA, as shown in figure 15*b*. This indicates that the new mortar has good bonding with the old mortar in the modified RAC.

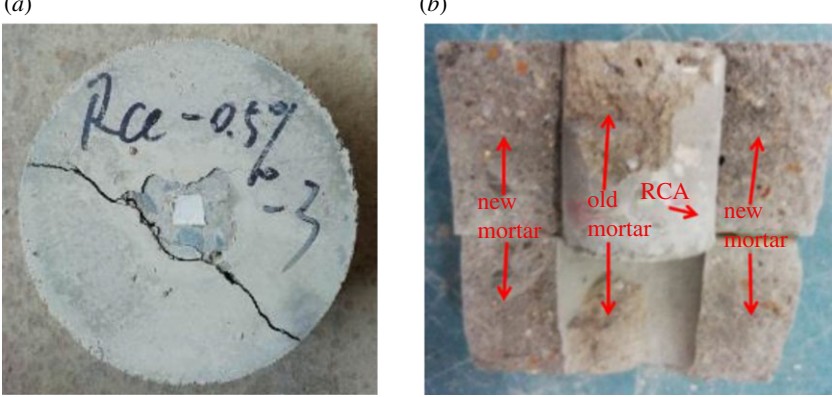

**Figure 15.** Failure pattern of the interface model specimens: (*a*) crack development and (*b*) splitting failure.

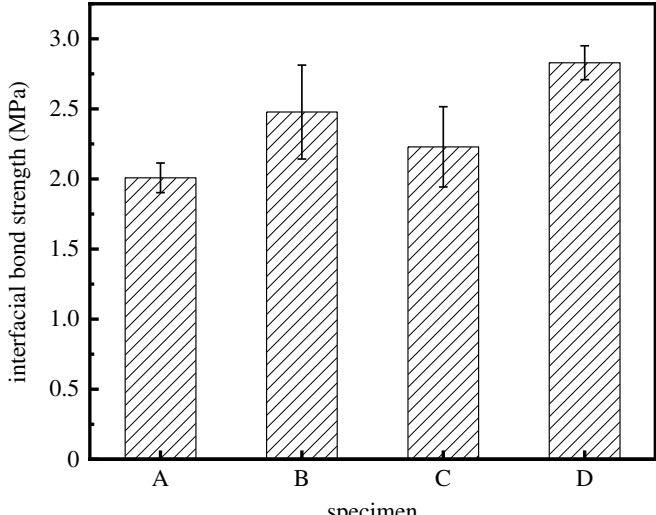

**Figure 16.** Bond strength of the ITZ model specimens.

### 3.2.2. Bond strength of the ITZ

The interfacial bond strength is the value of the maximum load, $P$, divided by the shear area (i.e. the total circumference and depth of the cylinder). The bond strength of interface model specimens is calculated as follows:

$$\tau = \frac{P_{\max}}{2\pi RL} \tag{3.1}$$

where $\tau$ is the bond strength of the ITZ (MPa), $P_{\max}$ is the maximum load (kN), $R$ is the radius of the cylindrical RCA (mm) and $L$ is the length of the RCA (mm).

The mean value of the interfacial bond strength is calculated and the results are shown in figure 16. Figure 16 shows that the interfacial bond strength of the untreated model specimens is 2.0 MPa (group A). Compared with the untreated model specimens, the interfacial bond strengths of the model specimens treated with nano-SiO$_2$ solution (group B), polyvinyl alcohol solution (group C) and replacing cement with fly ash (group D) are increased by 23.4%, 11% and 40.9%, respectively. Therefore, it can be concluded that each modification method can improve the interfacial bond strength between the RCA and new mortar in RAC. The optimal modification effect is achieved by replacing 10% of the cement with fly ash.

### 3.3. Discussion

On the one hand, with increasing replacement rate of RCA, the damping loss factor of the RAC beams increases. When the replacement rate reaches 100%, the loss factor of the RAC beams is 6.0% higher than

that of NAC. For improving the damping properties of RAC, the addition of the rubber powder is most effective. With the addition of 20% rubber powder, the damping capacity of the modified RAC is increased by 213.7% compared with the reference RAC. The loss factors of the RAC beams with steel fibres and fly ash increase little compared with that of the reference RAC.

On the other hand, with the increase in the loss factor, the strength of the RAC tends to decrease. When the coarse aggregate is completely replaced with RCA, the loss factor of the RAC is increased by 6.0%, while compressive strength of the RAC decreases by 22.6%. After adding 20% rubber powder, the damping capacity of the modified RAC is increased by 213.7%, while the compressive strength of the modified RAC decreases by 47.5%.

However, with the addition of steel fibre and fly ash, both the compressive strength and loss factor of the RAC specimens increase. With a steel fibre content of 10 wt%, the compressive strength and loss factor of the RAC increase by 21.9% and 15.2%, respectively. With a fly ash content of 25 wt%, the compressive strength and loss factor of the RAC increase by 8.6% and 6.9%, respectively. This indicates that the steel fibre and fly ash are effective for improving both the damping properties and compressive strength of RAC, and steel fibre is more effective than fly ash.

In this study, two methods were used for the modification of RAC: reinforcing the RCA through impregnation with a 0.5% polyvinyl alcohol emulsion and a nano-$SiO_2$ solution and strengthening the integrity of the RAC through the addition of fly ash as an admixture. The push-out test results for the ITZ model specimens show that replacing 10% of cement with fly ash in the new mortar is the best method for improving the interface strength.

Previous studies have also compared the effect of modification methods on the performance of RAC. Shaikh et al. [32] compared the effect of different modification methods on the ITZ performance of RAC. Pre-soaking of the RCA with a nano-$SiO_2$ solution and direct modification of the RAC with nano-$SiO_2$ as an admixture were considered. The results showed that the pre-soaking of RCA in the nano-$SiO_2$ solution provided more significant positive effects on the pore structure and strength of the RAC compared with the control RAC. However, the results reported by Li et al. [33] revealed that pre-mixing RCA with nanoparticles and integral strengthening of RAC by incorporating the nanoparticles did produce distinctly different effects on the compressive strength of RAC at 7 and 28 days. Further research is required on this topic.

## 4. Conclusion

A series of tests on the compressive strength, damping and ITZ performance of modified RAC are conducted, and the main conclusions are as follows:

(1) With increasing replacement rate of RCA, the damping loss factor of RAC blocks increases. When the replacement rate of RCA reaches 100%, the loss factor of the RAC is 6.0% higher than that of NAC; however, the compressive strength of the RAC decreases by 22.6%.

(2) After the addition of 20% rubber powder, the damping capacity of the modified RAC increased by 213.7%, while the compressive strength decreased by 47.5%. However, with the addition of steel fibre and fly ash, both the compressive strength and loss factor of the RAC specimens increase. With a steel fibre content of 10 wt%, the compressive strength and loss factor of the RAC increase by 21.9% and 15.2%, respectively. With a fly ash content of 25 wt%, the compressive strength and loss factor of the RAC increase by 8.6% and 6.9%, respectively. This indicates that steel fibre and fly ash are effective for improving both the damping properties and compressive strength, and steel fibre is more effective than fly ash.

(3) Two methods were used for the modification of RAC: reinforcing the RCA through impregnation with 0.5% polyvinyl alcohol emulsion or nano-$SiO_2$ solution and strengthening the RAC integrally through the addition of fly ash as an admixture. All these techniques can improve the interfacial bond strength between the RCA and new mortar. Replacing 10% of the cement with fly ash in the new mortar is the most effective method for improving the ITZ strength.

Data accessibility. Data relating to this work are provided in the electronic supplementary material.
Authors' contributions. B.L. designed the study, carried out the laboratory work and wrote the manuscript. H.L. and Z.T. carried out the analyses. Z.Y. conceived the study, designed the study, coordinated the study and helped draft the manuscript. All authors gave final approval for publication.
Competing interests. We have no competing interest.

Funding. This work was financially supported by the National Natural Science Foundation of China (grant no. 51668045), Jiangxi Science and Technology Committee (grant no. 20161BBG70056) and the Chinese Scholarship Council (grant nos. 201806825067, 201706955097).

Acknowledgements. The authors would like to express appreciation to Wu Jianpeng and Liu Zhecheng in School of Civil Engineering and Architecture (Nanchang University). We would also like to thank three anonymous reviewers for their helpful comments.

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
