## [Reviewer comments · Royal Society Open Science]

Review History

RSOS-190813.R0 (Original submission)

Review form: Reviewer 1

Is the manuscript scientifically sound in its present form?

Yes

Are the interpretations and conclusions justified by the results?

Yes

Is the language acceptable?

Yes

Do you have any ethical concerns with this paper?

No

Have you any concerns about statistical analyses in this paper?

No

Recommendation?

Accept with minor revision (please list in comments)

Comments to the Author(s)

This paper study on the compressive strength and damping properties of modified RAC with the addition of different proportions of recycled coarse aggregate, rubber powder, steel fiber, and fly ash are carried out. To elucidate the effect of the modification method on the ITZ performance of RAC, model ITZ specimens are used for push-out tests. This paper is generally well written, and it is recommended to be accepted for publication after the authors correct minor grammar errors and supplement relevant contents as follows:

1. In Table 6, "Specimen number" needs to be changed as "Specimen".
2. In Fig. 8, Fig. 10, Fig. 12, and Fig. 14, "Specimen number" needs to be changed as "Specimen".
3. In Fig. 16, the title of abscissa axis is not given.
4. In whole paper, "replacement ratio" should be changed with "substitution rate"
5. In whole paper, "replacement rate" should be changed with "substitution rate"
6. In Section 2.4., Please give more details about ITZ model specimens, such as the strength and diameter of cylindrical RCAs, and the diameter of ITZ model specimens.
7. There is a lack of relevant theoretical analysis in this paper, such as strength index conversion formula of recycled concrete and stress-strain relationship curves, etc. This content needs to be supplemented.

Review form: Reviewer 2

Is the manuscript scientifically sound in its present form?

Yes

Are the interpretations and conclusions justified by the results?

Yes

Is the language acceptable?

Yes

Do you have any ethical concerns with this paper?

No

Have you any concerns about statistical analyses in this paper?

No

Recommendation?

Accept as is

Comments to the Author(s)

This a well planned and executed study, result of which will be of interest to the field. I recommend the manuscript for publication in its current form.

Review form: Reviewer 3

Is the manuscript scientifically sound in its present form?

Yes

Are the interpretations and conclusions justified by the results?

Yes

Is the language acceptable?

Yes

Do you have any ethical concerns with this paper?

No

Have you any concerns about statistical analyses in this paper?

No

Recommendation?

Accept with minor revision (please list in comments)

Comments to the Author(s)

In general, the topic addressed in the paper "Experimental Study on the Compressive Strength, Damping, and ITZ Properties of Modified Recycled Aggregate Concrete" by B. Lei, et al. is very interesting. The paper shows novel results in the field of Recycled Aggregate Concrete through the use of rubber powder, steel fiber and fly ash materials. Meanwhile the paper needs minor changes in order to be suitable for publication in Royal Society Open Science.

Please find below a list of observations:

1. At the keywords: it is better to write "interfacial transition zone (ITZ)" instead of "ITZ".
2. Page 5, line 55: what was the size of corrugated steel fibers and fly ash materials? What kind of rubber was utilized? this have a special name? because this appears in the Table 3, with concentrations of five components.
3. Page 8 line 25: it says "...concrete cubes and have size of 100 mm x 100 mm x 100 mm and..." Please correct, "have size of 100 mm x 100 mm".
4. In the Table 6: it is included the values for compressive strength !!, but this is not mentioned at the title of the Table 6.

Decision letter (RSOS-190813.R0)

28-Oct-2019

Dear Dr Yao:

It is a pleasure to accept your manuscript entitled "Experimental Study on the Compressive Strength, Damping, and ITZ Properties of Modified Recycled Aggregate Concrete" in its current form for publication in Royal Society Open Science. The comments of the reviewer(s) who reviewed your manuscript are included at the foot of this letter.

Kind regards,
Anita Kristiansen
Editorial Coordinator
Royal Society Open Science
openscience@royalsociety.org

on behalf of Dr Ian Moore (Associate Editor) and Professor R. Kerry Rowe (Subject Editor).

Associate Editor Dr Ian Moore Comments to Author:

Comments to the Author:

The paper has been assessed by three expert reviewers. All indicate that it is deserving of publication, with two providing helpful suggestions regarding edits to the manuscript. None of these is critical, and so given the time taken in the review process, I recommend acceptance 'as is', but with encouragement for you to address the review comments in the final version they submit for copy editing (these edits will result in a better paper).

Reviewer(s)' Comments to Author:

Reviewer: 1

Comments to the Author(s)

This paper study on the compressive strength and damping properties of modified RAC with the addition of different proportions of recycled coarse aggregate, rubber powder, steel fiber, and fly ash are carried out. To elucidate the effect of the modification method on the ITZ performance of RAC, model ITZ specimens are used for push-out tests. This paper is generally well written, and it is recommended to be accepted for publication after the authors correct minor grammar errors and supplement relevant contents as follows:

1. In Table 6, "Specimen number" needs to be changed as "Specimen".
2. In Fig. 8, Fig. 10, Fig. 12, and Fig. 14, "Specimen number" needs to be changed as "Specimen".
3. In Fig. 16, the title of abscissa axis is not given.
4. In whole paper, "replacement ratio" should be changed with "substitution rate"
5. In whole paper, "replacement rate" should be changed with "substitution rate"
6. In Section 2.4., Please give more details about ITZ model specimens, such as the strength and diameter of cylindrical RCAs, and the diameter of ITZ model specimens.
7. There is a lack of relevant theoretical analysis in this paper, such as strength index conversion formula of recycled concrete and stress-strain relationship curves, etc. This content needs to be supplemented.

Reviewer: 2

Comments to the Author(s)

This a well planned and executed study, result of which will be of interest to the field. I recommend the manuscript for publication in its current form.

Reviewer: 3

Comments to the Author(s)

In general, the topic addressed in the paper "Experimental Study on the Compressive Strength, Damping, and ITZ Properties of Modified Recycled Aggregate Concrete" by B. Lei, et al. is very interesting. The paper shows novel results in the field of Recycled Aggregate Concrete through the use of rubber powder, steel fiber and fly ash materials. Meanwhile the paper needs minor changes in order to be suitable for publication in Royal Society Open Science.

Please find below a list of observations:

1. At the keywords: it is better to write "interfacial transition zone (ITZ)" instead of "ITZ".
2. Page 5, line 55: what was the size of corrugated steel fibers and fly ash materials? What kind of rubber was utilized? this have a special name? because this appears in the Table 3, with concentrations of five components.
3. Page 8 line 25: it says "...concrete cubes and have size of 100 mm x 100 mm x 100 mm and..." Please correct, "have size of 100 mm x 100 mm".
4. In the Table 6: it is included the values for compressive strength !!, but this is not mentioned at the title of the Table 6.
